# Enhancing Alginate Hydrogels as Possible Wound-Healing Patches: The Synergistic Impact of Reduced Graphene Oxide and Tannins on Mechanical and Adhesive Properties

**DOI:** 10.3390/polym16081081

**Published:** 2024-04-12

**Authors:** Sebastián Carrasco, Luisbel González, Mauricio Tapia, Bruno F. Urbano, Claudio Aguayo, Katherina Fernández

**Affiliations:** 1Laboratorio de Biomateriales, Departamento de Ingeniería Química, Facultad de Ingeniería, Universidad de Concepción, Concepción 4070386, Chile; scarrasco2017@udec.cl (S.C.); luisbgonzalez@udec.cl (L.G.); mtapia2017@udec.cl (M.T.); 2Departamento de Polímeros, Facultad de Ciencias Químicas, Universidad de Concepción, Concepción 3349001, Chile; burbano@udec.cl; 3Departamento de Bioquímica Clínica e Inmunología, Facultad de Farmacia, Universidad de Concepción, Concepción 4070112, Chile; caguayo@udec.cl

**Keywords:** alginate hydrogel, reduced graphene oxide, tannins, mechanical properties, adhesive properties

## Abstract

Hydrogels are three-dimensional crosslinked materials known for their ability to absorb water, exhibit high flexibility, their biodegradability and biocompatibility, and their ability to mimic properties of different tissues in the body. However, their application is limited by inherent deficiencies in their mechanical properties. To address this issue, reduced graphene oxide (rGO) and tannins (TA) were incorporated into alginate hydrogels (Alg) to evaluate the impact of the concentration of these nanomaterials on mechanical and adhesive, as well as cytotoxicity and wound-healing properties. Tensile mechanical tests demonstrated improvements in tensile strength, elastic modulus, and toughness upon the incorporation of rGO and TA. Additionally, the inclusion of these materials allowed for a greater energy dissipation during continuous charge–discharge cycles. However, the samples did not exhibit self-recovery under environmental conditions. Adhesion was evaluated on pig skin, revealing that higher concentrations of rGO led to enhanced adhesion, while the concentration of TA did not significantly affect this property. Moreover, adhesion remained consistent after 10 adhesion cycles, and the contact time before the separation between the material and the surface did not affect this property. The materials were not cytotoxic and promoted healing in human fibroblast-model cells. Thus, an Alg/rGO/TA hydrogel with enhanced mechanical, adhesive, and wound-healing properties was successfully developed.

## 1. Introduction

Hydrogels are three-dimensional crosslinked materials renowned for their ability to absorb water, exhibit great flexibility, their biodegradability and biocompatibility, and their ability to emulate the properties of diverse tissues in the body. This makes them ideal candidates for biomedical applications [1,2,3,4]. Various polymeric materials such as chitosan, collagen, or poly(vinyl alcohol) can be used in the formulation of hydrogels [5,6].

Alginate (Alg) is a natural polysaccharide made up of 1,4-linked β-D-mannuronate (M blocks) and β-L-guluronate (G blocks) units, capable of crosslinking through the addition of divalent or multivalent cations [7]. Alg is distinguished by its biocompatibility, non-toxicity, and biodegradable properties, rendering it suitable for use in tissue engineering, being capable of transmitting mechanical signals to cells or serving as a drug carrier, making it an ideal material for wound-healing dressings [8].

Despite its attributes, this biomaterial often exhibits low mechanical properties, which limits its use [9]. Since one of its roles is to provide mechanical integrity and also transmit mechanical signals to cells and tissues, this limitation is noteworthy [10]. Furthermore, the energy dissipation capacity of hydrogels contributes to their mechanical performance, and hydrogels with inadequate energy-dissipation efficiency tend to a have low resistance to crack propagation. Enhancing the ability to dissipate energy has been achieved through both covalent and non-covalent interactions, resulting in hydrogels that typically possess high tenacity [11]. The properties of Alg hydrogels under cyclic tensile stress have not been thoroughly investigated, despite the widespread use of cyclical mechanical characterization tests for analyzing these properties [12]. Similarly, studying the adhesion of these materials is crucial, given their potential application in wound dressings.

The mechanics of conventional wound dressings are not suitable for wounds, since when used, they have poor curative effects and limited lifespans. Moreover, when applied to stretchable areas of the body such as the ankle, elbow, knee, wrist, etc., the healing process becomes even more challenging. The proper adherence of the hydrogels is essential to ensure efficient healing in damaged biological tissues, and to prevent the dressings falling off, thereby exposing the wound [13]. Consequently, hydrogels must exhibit an appropriate mechanical strength to enhance their practicability, enabling them to adhere effectively to biological surfaces without causing damage during removal.

Recently, various techniques have been proposed to synthesize hydrogels with enhanced mechanical properties, including interpenetrating network hydrogels, double-network hydrogels, or nanocomposite hydrogels [14]. The incorporation of nanomaterials into hydrogels allows for improvements in mechanical properties by increasing crosslinking in the polymeric network, as well as enhancing adhesion to surfaces, thereby improving their potential use as wound dressings [15].

Graphene, a nanomaterial that consists of a 2D sheet with carbons with sp^2^ hybridization, possesses high mechanical resistance and remarkable electrical conductivity [16]. However, it is hydrophobic and poorly dispersible in biological media. In contrast, graphene oxide (GO), derived from graphene, is a graphene sheet functionalized with oxygenated groups, affording it greater dispersion than graphene. However, its mechanical properties, such as its elastic modulus, are lower than those of graphene [16]. The reduction of GO into reduced graphene oxide (rGO) allows for the partial restoration of graphene properties, presenting intermediate characteristics between graphene and GO, while still being dispersible in liquid media [17]. rGO can be incorporated into hydrophilic polymeric structures like hydrogels, involving various interactions such as π-π stacking, hydrogen bonding, or electrostatic interactions. These interactions result in materials with new physical and chemical properties, including high mechanical resistance.

Researchers have reported enhancements in the mechanical properties of agar, poly(acrylamide), or poly(vinyl alcohol) hydrogels by introducing rGO [18,19]. However, Alg hydrogels with rGO have been developed without evaluating their effect on mechanical properties [4,20,21].

The process of reducing GO can be accomplished through various chemical or thermal methods [22]. One approach involves the use of dopamine (DA) as a reducing agent [23], which undergoes autopolymerization under alkaline conditions [24], resulting in the formation of polydopamine (PDA)-coated rGO sheets [25]. Incorporating PDA into hydrogels has facilitated the development of hydrogels exhibiting strong adhesion to tissues [26,27], primarily attributed to the presence of catechol groups in the PDA [28].

While Alg hydrogels may potentially experience improvements through the incorporation of rGO, their effectiveness is not always guaranteed, often due to inefficient interactions between the polymer chains and the material. To address this challenge, the incorporation of a third compound, condensed tannins (TA), which exhibit a high possibility of interaction with diverse functional groups, can be considered a strategic junction point between the polymer and the nanocomposite. This aspect will be a consideration in the context of this proposal.

TA are water-soluble phenolic compounds found in various plant species known for their antioxidant and microbial activity [29,30,31], and have the ability to form complexes with polysaccharides and proteins. Additionally, non-covalent interactions may occur between the aromatic rings of tannins and aromatic species [32], such as π-π interactions, enabling phenolic groups of TA to bind to the rGO surface [33]. This interaction improves their mechanical properties [34]. On the other hand, Alg hydrogels with condensed TA derivatives have been developed [35], but a comprehensive mechanical characterization of these materials has not been conducted. Furthermore, the effect of incorporating both TA and rGO into Alg hydrogels has not been thoroughly studied.

Therefore, in the current study, Alg hydrogels were reinforced by incorporating escalating amounts of rGO and TA. The impact of these nanomaterials on the mechanical and adhesive properties of the resulting hydrogels was assessed, with the aim of their potential application in dermal or wound-healing scenarios. To validate these properties, the cytotoxicity and wound-healing potential of the materials were also evaluated.

## 2. Materials and Methods

### 2.1. Hydrogels Synthesis

The reduction of GO, previously synthesized using the Hummers method [36], was carried out with dopamine (DA) at reaction conditions of 60 °C for 24 h with a subsequent vacuum filtration and dialysis (MWCO = 6–8 kDa) process for 3 days (GO and rGO synthesis details are in Appendix A. The hydrogels’ synthesis was carried out in 42 mL of MilliQ water adding 4 mL of glycerol and 24 mg of FeCl_3_·6H_2_O with magnetic stirred to an ambient temperature. Then, rGO and TA were added in different mass proportions (Table 1). The solution formed was sonicated for 80 min in an ice bath, and then small amounts of Alg were added to obtain 1 g (1.9% *w*/*v*). Finally, borax (6 mL, 4% *w*/*w*) was added drop by drop until we obtained the hydrogel, and the mix was under agitation for 10 min to then be transferred to a mold and taken to the stove for 24 h at 50 °C. Nine hydrogels named Alg/rGOx/TAy were generated, where x and y represent mass proportions (0, 4.5, 9% *w*/*w*) of rGO and TA, respectively. The values of rGO and TA concentrations are the typically used when they are incorporated as additives in hydrogels [37,38].

### 2.2. Characterization of the Hydrogels

#### 2.2.1. Scanning Electron Microscopy (SEM)

The SEM analysis was used to investigate the micromorphology of GO and rGO. SEM images were recorded using a SEM VEGA3 SBU EasyProbe, TESCAN (Brno, Czech Republic) model microscope at 10 kV. The samples were coated using a gold-sputter coater and their surfaces were observed at different resolutions.

#### 2.2.2. Fourier Transform Infrared Spectroscopy (FTIR)

The FTIR was used to investigate the chemical nature of the interaction of the materials. The spectra were recorded in a Nicolet iS5 FTIR Spectrometer (Waltham, MA, USA). The wavenumber range analyzed was 4000–500 cm^−1^, and a total of 40 accumulated scans were acquired.

#### 2.2.3. X-ray Diffraction (XRD)

The X-ray diffraction (XRD) was used to determine the reduction degree of GO and the crystallinity of the hydrogels. The X-ray measurements were conducted using a Bruke Axs D4 Endeavor diffractometer (Bremen, Germany). The reference target used was Cu Kα radiation with a wavelength of 1.541841 Å and a power output of 2.2 kW. The voltage used was 40 kV, and the current was set at 20 mA. The samples were measured within a range of 2 to 50° for 141 s, with increments of 0.02°.

#### 2.2.4. Mechanical Characterization

The mechanical properties of the Alg/rGO/TA composites were measured using a Universal Testing Machine Shimadzu EZ-XS (Kyoto, Japan) equipped with a 20 N load cell. All samples were cut following the dumbbell template of 10 mm width, 55 mm length, and 1 mm thickness. The samples were held between two clamps and pulled by the top clamp at a velocity of 5 mm/min. The elongation and breaking force were measured when the hydrogels tore apart. The elongation at break (*ε*) and tensile strength (*σ*) were calculated using Equations (1) and (2), were ΔL is the deformation of the sample, Lo the initial sample length, $P_{max}$ the breaking force, and Ao the cross-sectional area of the sample.
(1)$$\varepsilon \;\left(\%\right)=\frac{\Delta L(mm)}{Lo(mm)}\times 100\%$$
(2)$$\sigma \;\left(kPa\right)=\frac{P_{max}(kN)}{Ao\;(m^{2})}$$

Also, the elastic modulus was calculated using Equation (3). Tenacity corresponds to the area-under-the-force displacement curve between the initial point and the breaking point; Equation (4):(3)$$Elastic\;modulus\;(kPa)=Slope\times \frac{Lo(mm)}{Ao\;({mm}^{2})}$$
(4)$$Tenacity\;(kJ/m^{3})=\int_{0}^{\varepsilon_{max}}\sigma d\varepsilon$$

For hysteresis measurement, the samples were first stretched to a deformation of 10% at 5 mm/min, and then unloaded at similar speeds and repeated for 10 cycles of each sample. The dissipated energy, which is defined as the area of the hysteresis loop encompassed by the loading–unloading curve, is calculated by integrating the area between loading–unloading curves using Equation (5).
(5)$$Hysteresis\;(kJ/m^{3})=\int_{0}^{\varepsilon_{max,\;loading}}\sigma d\varepsilon -\int_{0}^{\varepsilon_{max,\;unloading}}\sigma d\varepsilon$$

The self-recovery ability of hydrogels was investigated by using cyclic tensile measurements. The sample was initially stretched at a strain of 10% and then unloaded. After each loading–unloading cycle, the sample was relaxed for certain amount of time (0, 5, 15 and 30 min) and processed to the next cycle. The recovery was defined as the ratio of energy dissipation after a certain relaxed time with respect to the initial energy dissipation. Before mechanical tests, all the samples where maintained at 20 °C and 50% relative humidity.

#### 2.2.5. Adhesive Properties

The tissue adhesiveness of the hydrogels was characterized by a tensile-adhesion test using porcine skin to mimic the natural tissue following the assembly of Figure 1a on a Universal Testing Machine Shimadzu EZ-XS (Kyoto, Japan) equipped with a 20 N load cell (Figure 1b). Porcine skin was used as the represented skin tissue. The porcine skin was attached to a glass slide with cyanoacrylate glue. The hydrogels were adhered to the porcine skin with a bonded area of 15 mm × 15 mm and sandwiched by the other piece of porcine skin for 1 min with a constant force of 200 g_f_. Then, the samples were pulled to failure with a crosshead speed of 1 mm/min. The adhesion strength was calculated by the maximum stress divided by the bonded area. Also, the adhesion strength was evaluated after 10 cycles of adhesion. In addition, the effect of contact time on adhesion prior to separation was evaluated considering times of 10, 30, 60, and 300 s.

#### 2.2.6. Cytotoxicity Assay

Cytotoxicity was assessed by conducting the 3-(4,5-dimethylthiazol-2-yl)-2,5-diphenyltetrazolium bromide (MTT) assay on human dermal fibroblast (HDF) cell lines (Sigma Aldrich, Santiago, Chile), which were obtained from adult human epithelial tissue (Sigma, Chile). The hydrogels were cut into circular discs (28 mg, 3.4 ± 0.6 mm diameter) and together with the individual materials (Alg and rGO) were subjected to UV radiation for 30 min. After sterilization, materials were added to a 6-well plate containing 1 mL of DMEM (Dulbecco’s Modification Eagle’s Medium) for every 10 mg of material added. The materials were incubated for 24 h at 37 °C and the liquid extract was filtered using a 0.22 μm cellulose acetate filter (STARLAB, Milton Keynes, United Kingdom). Then, it was mixed with 5% (*v*/*v*) fetal bovine serum (FBS) and 1% (*v*/*v*) antibiotics (100 µ/mL of penicillin and 100 µ/mL of streptomycin). The cell density of the HDFs was adjusted to 104 cells per well and they were cultured in DMEM nutrient medium in 96-well plates at 37 °C with a 5% CO_2_ atmosphere until a monolayer was formed. Subsequently, the nutrient medium was removed from the wells, and 100 μL of the material extracts were added to each well. After 48 h of incubation at 37 °C in a 5% CO_2_ atmosphere, 20 μL of MTT (5 mg/mL) were added to each well and incubated for an additional 4 h. To dissolve the formazan crystals, 100 μL/well of DMSO were added. The absorbance was determined using a microplate reader (Biotek synergy 2, Agilent Technologies, Santa Clara, CA, USA) at a wavelength of 540 nm. The relative cell viability was calculated using Equation (6):(6)$$Relative\;cell\;viability\;(\%)=\frac{A_{540}\;of\;treated\;cells}{A_{540}\;of\;control\;cells}\times 100$$

#### 2.2.7. In Vitro Wound-healing Assay (Scratch Test)

The healing assays were carried out using HDF cells, with 5 × 10^4^ cells/well being seeded in 24-well plates. The plates were supplemented with a DMEN medium containing 10% SFB and 1% antibiotic. The cells were cultured at 37 °C in a CO_2_ atmosphere of 5% until they reached 100% confluence. Then, the HDF monolayer was rinsed with PBS and a scratch in the center was created manually using a sterile plastic tip. Following the mapping of the wounds, they were thoroughly washed with PBS to ensure the removal of any detached cells. Subsequently, each material sample was fixed within the CellCrown 24 inserts (Corning Incorporated, Pittston, PA, USA) and carefully placed in the designated wells of a 24-well plate without any contact with the surface. The progress of wound closure was observed at 12 h intervals using a light microscope (MOTIC AE31, Richmond, BC, Canada) for up to 48 h. Finally, the images were analyzed using ImageJ 1.53e^®^ software (National Institutes of Health, Bethesda, MD, USA). The wound-closure rates were calculated according to Equation (7):(7)$$Rate\;of\;wound\;closure\;(\%)=\frac{A_{0}-A_{t}}{A_{0}}\times 100$$
where *A*_0_ is the initial wound area and *A_t_* is the wound area after each time interval.

### 2.3. Statistical Analysis

Each experiment was carried out three times, and the measurements were performed in triplicate. Data analysis was performed using OriginPro8.5^®^ software (OriginLab Corporation, Northampton, MA, USA). The analysis of variances (ANOVA) with an accepted significance of *p*-value < 0.05 and the multiple range analysis (Tukey’s test, 95% confidence) were performed in Statgraphics Centurion XVII^®^ software (Statgraphics Technologies, Inc., The Plains, VA, USA). Data are presented as means ± SD, and the error bars are shown in each figure.

## 3. Results and Discussion

The materials were synthesized, and their morphological, chemical, mechanical, and biological characteristics were subsequently analyzed. The obtained results are described below.

### 3.1. Morphological Studies

The GO morphology was examined through SEM and compared with rGO topography. In Appendix A, the GO surface is depicted, displaying wrinkled scales as result of the exfoliation process following graphite oxidation [39]. Additionally, surface unevenness is evident, attributed to the presence of multiple lateral layers. GO was then reduced by PDA to obtain rGO (Appendix A), reveling a number of fine wrinkled sheets on the surface. This observation indicates a distinct structure from GO, marked by the formation of irregular, foldable, and disordered layers, suggesting that PDA covers the GO [40].

Alg presented a regular and homogenous surface (Appendix A). The rGO inclusion modified the smooth apparency of Alg, resulting in noticeable changes in the morphological aspect of hydrogels, particularly with increasing rGO concentration (Figure 2a vs. Figure 2b). The rough surface is associated with the incorporation of rGO sheets into the matrix in a disperse phase. The addition of TA to Alg produced a homogenous surface (Figure 2c), potentially caused by molecular interactions among the multiactive sites of both compounds. When the hydrogel was formed with all materials, Alg/rGO_4.5_/TA_9_ (Figure 2d), the surface displayed roughness with wrinkles, resembling the neat surface of rGO but covered by a film. Similar observations were made for when creating a hydrogel of Alg with 10% rGO, observing adjacent layers connected to each other [21].

Liu et al. studied the incorporation of catechin and acid tannins into Alg in different proportions (0.5%, 2% and 5%) and found that the incorporation formed an uniform crosslinked network, especially with acid tannins [41]. Despite the similar name, acid and condensed tannins differ in their chemical structure, molecular weight, and reactivity [42]. In the material created here, condensed tannins (TA) with molecular weights presented in Appendix A and a negative surface charge of −48.5 mV were incorporated to the hydrogel in two proportions (4.5 and 9% *w*/*w*) making an homogeneous hydrogel. Facchi et al., [35] made an hydrogel of alginate and Tanfloc (a cationic biopolymer obtained from natural condensed tannins), and they observed large non-homogeneous pores and fragile morphologies. Thus, the interaction among Alg/rGO/TA produced a crosslinked structure, different to that previously reported.

### 3.2. Chemical Characterization of the Hydrogels

The pure materials and the hydrogels were analyzed by FTIR to confirm the presence of functional groups (Figure 3a and Figure 3b, respectively). The GO spectrum (Figure 3a) shows vibrational bands at 3335 cm^−1^ (O–H), 1730 cm^−1^ (C=O), 1635 cm^−1^ (C=C), 1194 cm^−1^ (C–O–C), and 1050 cm^−1^ (C–O), confirming the oxidation of graphite. The rGO spectrum shows a decrease in the intensity of bands related to carbonyl, alkoxy, and hydroxyl groups. Additionally, there is the emergence of the amide group band (1546 cm^−1^) [43]. This change preliminarily verifies DA incorporation into GO through the formation of amide bonds between GO oxygen groups and DA amine groups, in agreement with previous studies [44,45,46]. DA exhibits bands at 1619 and 1500 cm^−1^, corresponding to the C=C stretching vibration of the benzene ring and the −OH stretching vibration of the DA diols [47]. The TA spectrum reveals the presence of −OH groups in the range of 3550–3100 cm^−1^, along with a characteristic band of flavonoids at 1280 cm^−1^ for TA from pine [48].

Figure 3b illustrates the spectrum of Alg/rGO/TA hydrogels. The bands at 3340 cm^−1^ and 2919–2885 cm^−1^ correspond to the stretching vibration of the O–H and C–H bonds, respectively. Furthermore, the antisymmetric and symmetric vibration of the carboxylic group is observed around the 1650–1610 cm^−1^ and 1414 cm^−1^ peaks, respectively [49]. The signals between 1300 and 1000 cm^−1^ can be associated with the stretching characteristics of C–C–H, O–C–H and C–O–C.

The XRD patterns of the hydrogels are presented in Figure 3c and the interplanar distance in Appendix A. The GO pattern revealed a very sharp diffraction peak at 2θ = 8.65°, with an interlayer distance of 10.22 Å. These features confirm the oxidation of graphite and the intercalation of oxide functional groups on the carbon basal plane, such as epoxy, hydroxyl, carbonyl, and carboxyl groups. After the GO reduction with DA, a broad peak at 2θ = 21.72° was observed, reducing the interplanar distance to 4.09 Å. This reduction may be attributed to the π-π stacking interactions of the hexagonal cells of graphene and the DA aromatic ring, indicating a successful GO reduction [50].

Alg and the hydrogels presented a single broad peak, evidencing a non-typical crystal shape. Alg and Alg with the addition of rGO or TA showed similar XRD patters, (2θ = 21.2°, d 4.19 vs. 2θ = 21.5°, d 4.13 Å). However, with the addition of both fillers (Alg/rGO/TA_9_), the amorphous structure of the samples increased (2θ = 21.7°; 4.10 Å). The hydrogen bonds among three compounds could hinder the molecular directional arrangement of each polymer, promoting a larger amorphous region in this sample. These results align with previous studies on a hydrogel of Gel/Alg/catechin and Gel/Alg/tannin acid, where increasing amounts of catechin and tannin acid were added to the blend, resulting in a broad peak, also with a value of 2θ = 21.5° [41].

### 3.3. Mechanical Characterization of the Hydrogels

Figure 4 illustrates the impact of incorporating different amounts of rGO and TA into Alg hydrogels in tensile tests. The tensile stress–strain curves are presented in Appendix A, as the mean of three determinations. By utilizing these conditions, one can assess the quality of a material, estimate its performance in various cargo scenarios, and craft structural elements accordingly. Upon analysis, the unreinforced Alg hydrogel presented a tensile strength of 84.9 kPa, a maximum deformation of 63.8%, an elastic modulus of 169.2 kPa, and a toughness of 22.4 kJ/m^3^. These values fall within the ranges reported in previous studies characterizing the mechanical properties of Alg hydrogels [9].

In Figure 4a, it is evident that incorporating rGO into the Alg hydrogel (Alg/rGO_9_) significantly improves tensile strength, reaching a value of 168.8 kPa. Similarly, the introduction of TA alone to the Alg matrix also increases tensile strength to 179.4 kPa (Alg/TA_9_). The maximum elongation of the materials was not affected by the rGO addition (Figure 4b), but TA incorporation did impact it, increasing the maximum elongation. This is likely caused by a close interaction between the components, resulting in a homogeneous pattern, as observed in the SEM images (Appendix A vs. Figure 4c). According to Figure 4c, the addition of rGO to Alg hydrogels proportionally improved the elastic modulus, reaching values of 259 kPa and 412 kPa for Alg/rGO_4.5_ and Alg/rGO_9_ hydrogels, respectively. The TA addition to Alg hydrogels, in increasing amounts, also enhances the elastic modulus (Alg/TA_4.5_ and Alg/TA_9_). Figure 4d shows that the addition of rGO improves the toughness of Alg hydrogels (Alg/rGO_4.5_ and Alg/rGO_9_ hydrogels), and the TA addition also enhances this property (22.4 for Alg to 58.6 kJ/m^3^ for Alg/TA_4.5_ and 72.4 kJ/m^3^ for Alg/TA_9_). The combination of rGO and TA in Alg/rGO_4.5_/TA_9_ also follows the trend of increasing mechanical properties with respect to Alg, with tensile strength increased by 101%, elastic modulus by 83%, and toughness by 187%. Thus, the introduction of rGO and TA into Alg hydrogels allows for an improvement in mechanical properties, with the Alg/rGO_4.5_/TA_9_ hydrogel presenting the most remarkable mechanical behavior in tensile tests. Finally, comparing the effect of rGO on its own, TA on its own, and the combination of rGO and TA, the samples that presented a higher mechanical resistance with respect to Alg were Alg/rGO_9_, Alg/TA_9_, and Alg/rGO_4.5_/TA_9_. In Table 2, the mechanical properties of these samples are summarized.

Thus, Alg hydrogels with increasing rGO concentration exhibit improved mechanical properties, attributed to the hydrogen-bonding interactions of hydroxyl groups and high interfacial adhesions between rGO and Alg chains [51]. Hydrogen-bonding interactions among the components could also explain the enhancements in TA-only Alg hydrogels. The mechanical response in Alg/rGO/TA hydrogels may be influenced by possible agglomerations and irregularities of the nanomaterials within the matrix or shielding effects between them, as was observed in SEM images (Figure 2d and Appendix A). For example, in the Alg/rGOx/TA_4.5_ samples (Appendix A), an increase in rGO concentration can lead to the agglomeration of the nanomaterial together with TA, making the material more brittle. On the other hand, in the Alg/rGOx/TA_9_ samples (Figure 4 and Appendix A), the reduction in mechanical properties was not as pronounced as observed in Alg/rGOx/TA_4.5_. This difference could be attributed to the higher TA available for interaction with rGO. The evaluation of mechanical properties by tensile tests of hydrogels based on natural biopolymers including rGO is scarce, since synthetic polymers are generally used for these assessments. Additionally, the TA effect and its combination with rGO on Alg hydrogels has not been reported, making this finding a novel result.

#### 3.3.1. Hysteresis and Self-Recovery of the Hydrogels

This study of hysteresis and self-recovery aimed to analyze the response of the material under repeated utilization and detect damage. The hysteresis of the hydrogels was evaluated through charge and discharge cycles, providing a measure of the material’s recovery capacity or its energy dissipation [52]. The cyclic tensile curves after 1, 2, and 10 load–unload cycles of the materials are presented in Appendix A, reveling hysteresis in all the curves.

A comparison of the hysteresis values of the stress–strain curves of the different materials is presented in Figure 5a. The Alg energy dissipation after one charge–discharge cycle was 0.117 kJ/m^3^, a value that remains almost unchanged after 10 cycles. After one charge–discharge cycle, the TA introduction produced an increase in hysteresis in the Alg/rGO_9_ hydrogel (0.226 kJ/m^3^). The incorporation of rGO produced a significant increase in the hysteresis of the Alg/TA_9_ hydrogel (0.321 kJ/m^3^). Finally, the Alg/rGO_4.5_/TA_9_ hydrogel shows remarkable energy dissipation (0.380 kJ/m^3^), reaching the highest hysteresis value among all samples. When evaluating the effect of rGO and TA on the hysteresis of hydrogels after multiple charge–discharge cycles, it is evident that energy dissipation decreases as the number of cycles increases in hydrogels composed of Alg/rGO_9_, Alg/TA_9_, and Alg/rGO_4.5_/TA_9_. The results suggest that the greater dissipation of energy in the Alg/rGO_9_, Alg/TA_9_, and Alg/rGO_4.5_/TA_9_ hydrogels could be caused by the chain dissociation formed by non-covalent interactions, such as hydrogen bonds [53], between rGO and TA with the polymeric chains of Alg. On the other hand, the decrease in energy dissipated after multiple charge–discharge cycles can be attributed to the fact that the hydrogel network was irreversibly altered after the first cycle.

The self-recovery capacity of the hydrogels at room temperature was evaluated through cyclic charge–discharge tests with recovery times between each cycle (0, 5, 15, and 30 min) (Figure 5b). Here, the recovery corresponds to the area ratio of the hysteresis loops with respect to the first cycle. Without considering a recovery time (t = 0), the Alg hydrogel dissipated 73.5% of the initial energy. However, an increase in recovery time did not improve the dissipated energy capacity, which is probably because most of the interactions are irreversibly broken. The Alg/rGO_9_ hydrogel presented 67.9% of the initial hysteresis without considering recovery time. However, if recovery times of 5 min or 15 min are considered, 75.5% and 74.9% of the initial hysteresis were reached, respectively. The Alg/TA_9_ hydrogel reached 73.3% of the initial hysteresis without considering recovery time. After 5 min of recovery, 85.7% of the initially dissipated energy was reached; the highest value among all the samples. The Alg/rGO_4.5_/TA_9_ hydrogel dissipated 71.6% of the initial energy without recovery time, a value that increases to 79% when considering 5 min of recovery. After 30 min of recovery, the samples Alg/rGO_9_, Alg/TA_9_, and Alg/rGO_4.5_/TA_9_ presented the same energy dissipation as when recovery time was not considered. Those results suggests that there is a part of the structure that is capable of self-recovery; however, there is another part that breaks irreversibly and is not capable of self-repair under room-temperature conditions, as is evidenced in Alg hydrogels. The introduction of rGO and TA within the matrix (samples Alg/rGO_9_, Alg/TA_9_ and Alg/rGO_4.5_/TA_9_) results in a greater material recovery than Alg, caused by possible interactions that occur between these materials and the alginate chains.

#### 3.3.2. Adhesive Properties of the Hydrogels

The adhesive properties of the hydrogels were studied on pig skin (Figure 1) due to its similarity to human skin [26]. Figure 6a shows the adhesion strength of the hydrogels on this substrate, being 1.75 kPa for Alg. The rGO incorporation into the matrix produced the highest adhesion force values, reaching 7.18 kPa and 4.33 kPa for Alg/rGO_4.5_ and Alg/rGO_9_, respectively. This is mainly attributed to the PDA-rGO complex present in the hydrogels and the adhesive characteristics of the PDA groups’ catechol, allowing for binding with other active groups of the alginate [28]. The decrease in the adhesion force at a higher concentration of rGO may be due to an agglomeration of PDA-rGO that produces inefficiencies in the adhesive force. On the other hand, the TA addition to these samples produces a decrease in adhesive force, possibly due to a shielding effect on the PDA-rGO groups, as observed in SEM images. Previous studied have examined the adhesiveness of hydrogels and the effect of the use of dopamine. A hydrogel has potential use in the wound healing of polyvinyl acetate, sodium alginate, and tannic acid, achieving an adhesion strength on pig skin of 13.1 kP [54]. An oxidized sodium alginate, dopamine, and polyacrylamide hydrogel that achieved an adhesion force on pig skin of 6.5 kPa, remaining constant even after three adhesion cycles [55]. These values tend to be slightly higher than those obtained in this investigation, possibly due to the different formulation of the hydrogels. Research findings suggest that a bandage possessing an adhesiveness ranging from 5 to 10 kPa has been deemed suitable for wound-healing purposes. This adhesive strength is carefully calibrated to ensure that it can be safely removed from the regenerated area without causing any harm or damage [26].

The effect of the repetitive use of the material is presented in Figure 6b, where it is observed that after a period of use of 10 times, there are no significant differences in the adhesion force. This behavior is similar for all the samples. If the objective is to use the patch as a wound-healing device, these results are significant since this property allows it to be used several times without losing its adhesion capacity. Figure 6c shows the effect of the contact time between the material and the surface of the substrate prior to separation for the Alg, Alg/rGO_4.5_, and Alg/rGO_9_ samples, which did not show significant differences. The results suggest that the adhesive properties are mainly due to the presence of the PDA-rGO complex in the materials and that the repetitive use of the material together with the contact time prior to separation do not significantly influence the adhesive strength on pig skin.

### 3.4. Biological Characterization

#### 3.4.1. Cytotoxicity Assays

A wound dressing must be biocompatible, non-toxic, and support the process of cell migration [56]. The biocompatibility of all synthetized materials was evaluated, and values above 80% were observed in all cases (Figure 7a). The rGO addition to Alg had a favorable effect, especially for the Alg/rGO_4.5_ sample. Previous studies have demonstrated that incorporating graphene derivatives at low concentrations into Alg materials enhances cell viability and provides a protective effect on biological activity in vitro [57]. Therefore, the use of low concentrations of rGO in the materials favors cell survival and stimulates cell growth [58].

The incorporation of TA into the materials showed a reduction in the cell viability from 142.6% Alg/rGO_4.5_ to 86.5% for Alg/rGO_4.5_/TA_4.5_ and 101.5% for Alg/rGO_4.5_/TA_9_. This phenomenon could be attributed to the deterioration of the polyelectrolytic surface with the TA incorporation. Hydrogels containing 9% rGO showed a linear increase in cell viability with increasing TA concentration. In summary, all hydrogels showed a rise in cell viability when TA concentration increased. The findings are similar to that reported by Jafari et al., who concluded that this phenomenon can be attributed to the enhanced hydrophobicity of the hydrogels after being treated with TA [59]. Thus, the hydrogels synthesized were non-toxic for human dermal fibroblasts.

#### 3.4.2. In Vitro Wound-healing Assay (Scratch Test)

Fibroblasts play a crucial role in the tissue repair process, due to their ability to migrate and proliferate [60]. The ability of cell migration was evaluated by measuring the closure of a wound created in a fully confluent monolayer of fibroblast as an in vitro wound-healing model. Other authors have also used this model to assess the performance of hydrogels as patches [61,62].

Visually (Figure 7b), TA addition had a more favorable effect on cell migration than rGO alone at the evaluated concentrations, and the Alg/rGO_4.5_/TA_4.5_ sample maintained a more favorable HDF migration compared to Alg or other samples with only rGO at 48 h, which was significant. It suggests that the combination of rGO and TA in the Alg hydrogel could have a synergistic effect, being potential candidates in the wound-healing process. In terms of wound-closure rate, where more times are displayed (Figure 7c), it can be observed that rGO inclusion had a favorable effect on the migration compared with Alg, but increasing the amount of rGO was not as beneficial, similarly to cell viability results. TA incorporation into the hydrogels showed a positive effect in all cases studied. Thus, it can be concluded that TA addition has no negative impact and enhances the viability and migration ability of fibroblasts, ultimately promoting cell proliferation. The positive effects of TA inclusion in alginate hydrogels have been supported by previous research. It has been suggested that the formation of hydrogen bonds between the catechol group of TA and the thiols or imidazoles present in the cytomembrane of fibroblasts may be responsible for these effects [63].

## 4. Conclusions

Alg/rGO/TA hydrogels with potential uses as wound dressings were developed by evaluating the effects of rGO and TA concentration. The mechanical properties of the Alg hydrogel were improved by incorporating rGO and TA independently and in combination. The resistance of the hydrogels in cyclic tensile tests was also superior to that of the Alg hydrogel. The adhesiveness of the hydrogels was mainly enhanced by rGO incorporation due to the presence of the PDA-rGO complex formed after GO reduction. The materials maintain their tackiness after repeated use, and the contact time prior to separation did not impact this property. The hydrogels were not cytotoxic towards human dermal fibroblasts, while the Alg/rGO_4.5_ material displayed a remarkable cell viability of 142.6%. Furthermore, the TA inclusion not only boosted cell viability but also facilitated the migration of fibroblasts. These findings indicate the synergistic effect of incorporating both rGO and TA in the in vitro experiments, highlighting the hydrogels’ potential as possible wound patches.

## Figures and Tables

**Figure 1 polymers-16-01081-f001:**
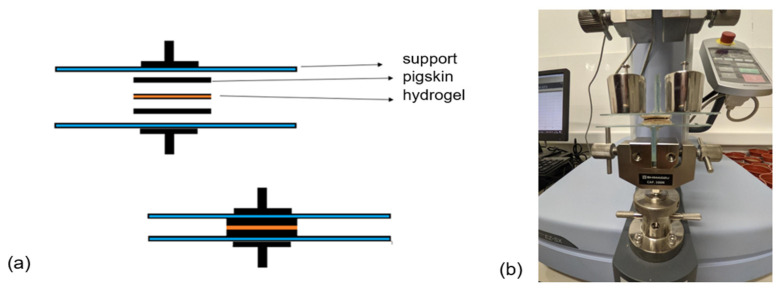
(**a**) Schematic representation for adhesive test and (**b**) prior assembly performing adhesion test.

**Figure 2 polymers-16-01081-f002:**
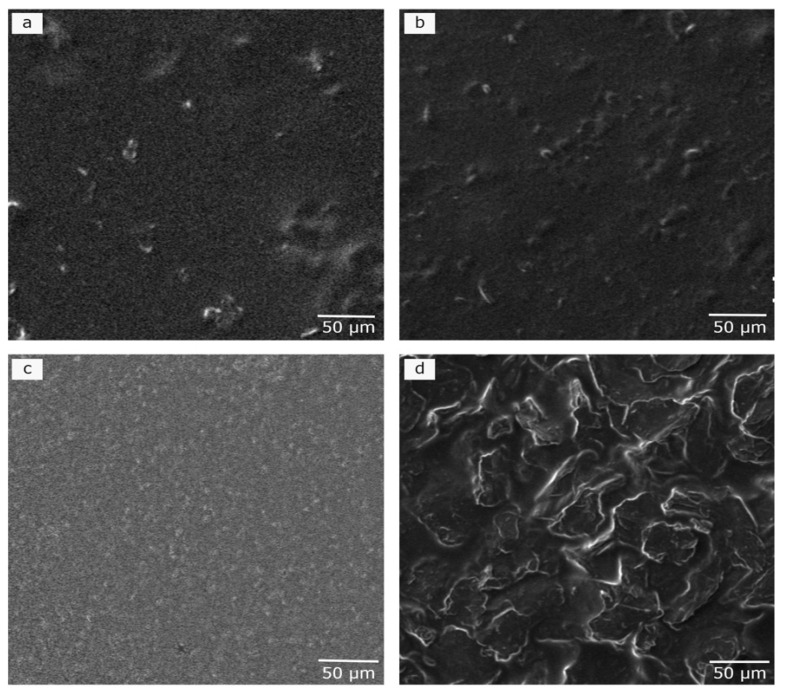
Scanning electron microscopy (SEM) images of (**a**) Alg/rGO_4.5_, (**b**) Alg/rGO_9_, (**c**) Alg/TA_9_, and (**d**) Alg/rGO_4.5_/TA_9_.

**Figure 3 polymers-16-01081-f003:**
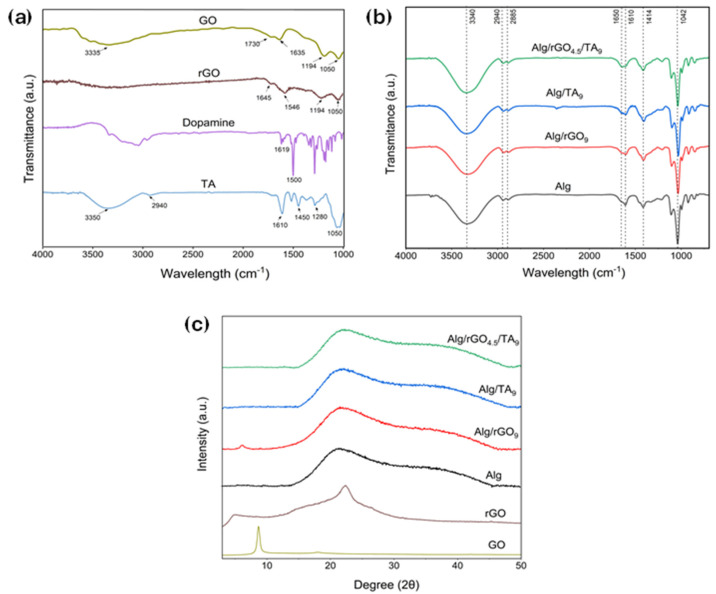
(**a**) FTIR spectra of GO, rGO, DA, and TA; (**b**) FTIR spectra of Alg, Alg/rGO_9_, Alg/TA_9_, and Alg/rGO_4.5_/TA_9_; (**c**) XRD spectra of GO, rGO, Alg, Alg/rGO_9_, Alg/TA_9_, and Alg/rGO_4.5_/TA_9_.

**Figure 4 polymers-16-01081-f004:**
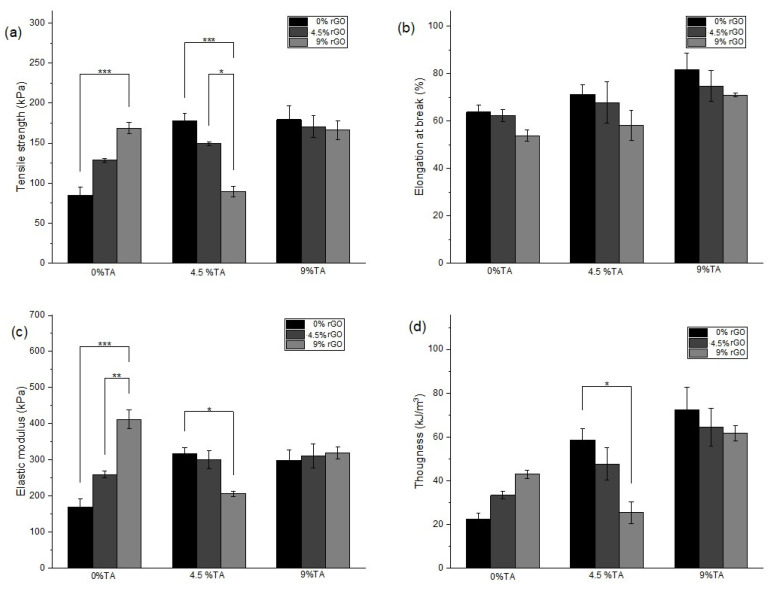
Mechanical properties in tensile tests including (**a**) tensile strength, (**b**) elongation at break, (**c**) elastic modulus, and (**d**) toughness for Alg hydrogels with different rGO and TA contents. (*) *p* ≤ 0.05 (**) *p* ≤ 0.01 and (***) *p* ≤ 0.001.

**Figure 5 polymers-16-01081-f005:**
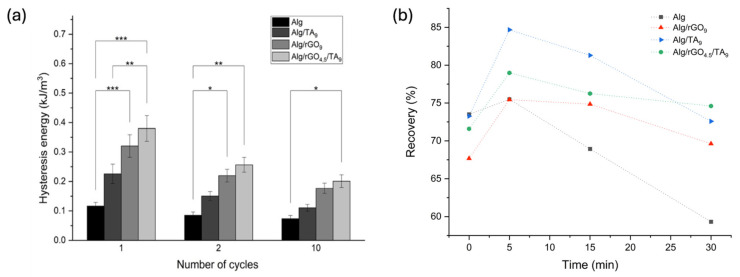
(**a**) Hysteresis properties of Alg hydrogels with different rGO and TA contents. (**b**) Self-recovery properties of Alg hydrogels with different rGO and TA contents. (*) *p* ≤ 0.05 (**) *p* ≤ 0.01 and (***) *p* ≤ 0.001.

**Figure 6 polymers-16-01081-f006:**
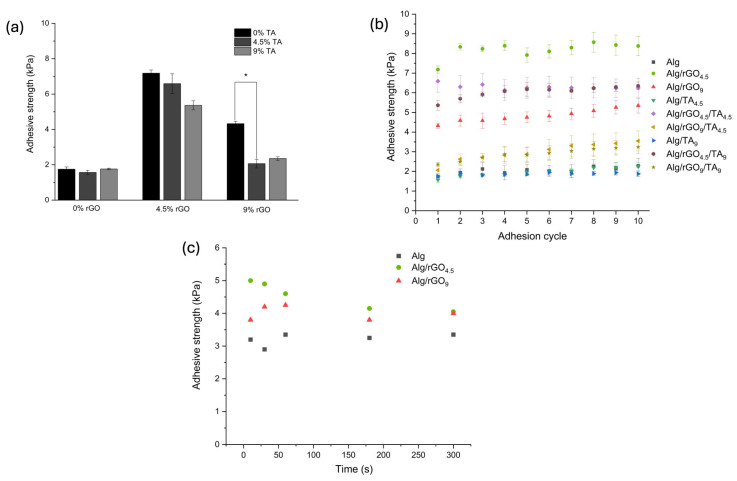
Adhesive properties of hydrogels with different rGO and TA contents in pigskin, including (**a**) the corresponding adhesion strength, (**b**) repeated adhesion of the hydrogels to the substrate, and (**c**) the effect of hydrogel–substrate contact time prior to separation. (*) *p* ≤ 0.05.

**Figure 7 polymers-16-01081-f007:**
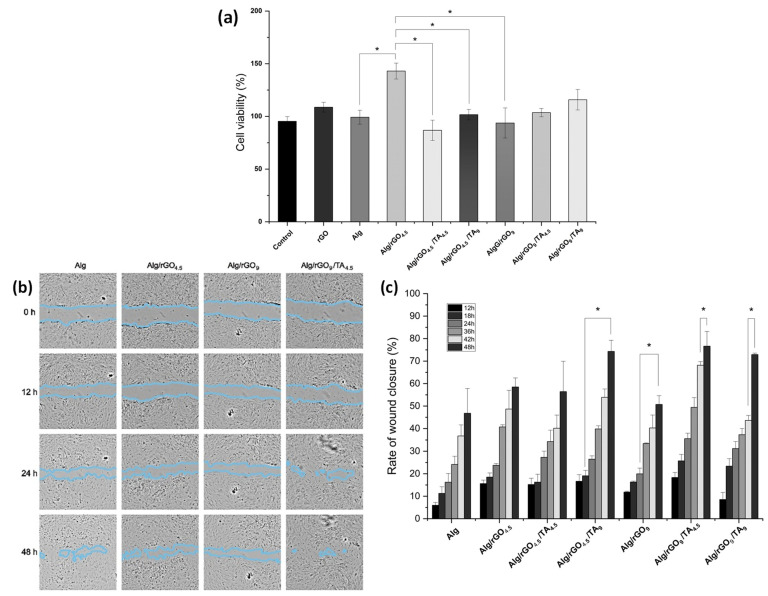
(**a**) Cell viability of human dermal fibroblasts in the presence of rGO, Alg, Alg/rGO_4.5_, Alg/rGO_4.5_/TA_4.5_, Alg/rGO_4.5_/TA_9_, Alg/rGO_9_, Alg/rGO_9_/TA_4.5_, and Alg/rGO_9_/TA_9_. (**b**) Cell migration in wound closure as a function of time and (**c**) rate of wound closure (%) for Alg, Alg/rGO_4.5_, Alg/rGO_4.5_/TA_4.5_, Alg/rGO_4.5_/TA_9_, Alg/rGO_9_, Alg/rGO_9_/TA_4.5_, and Alg/rGO_9_/TA_9_. The asterisk indicates significant differences at a 95% confidence level based on Tukey’s test.

**Table 1 polymers-16-01081-t001:** Composition and nomenclature of hydrogels.

Nomenclature	0% rGO	4.5% rGO	9% rGO
0% TA	Alg	Alg/rGO_4.5_	Alg/rGO_9_
4.5% TA	Alg/TA_4.5_	Alg/rGO_4.5_/TA_4.5_	Alg/rGO_9_/TA_4.5_
9% TA	Alg/TA_9_	Alg/rGO_4.5_/TA_9_	Alg/rGO_9_/TA_9_

The percentages used (*w*/*w*) are correlated with the mass of the alginate.

**Table 2 polymers-16-01081-t002:** Mechanical properties of Alg hydrogels with different rGO and TA contents.

Hydrogels	Tensile Strength (kPa)	Elongation (%)	Elastic Modulus (kPa)	Toughness (kJ/m^3^)
Alg	84.9 ± 17.6 ^a^	63.8 ± 4.9 ^a^	169.2 ± 41.5 ^a^	22.4 ± 4.5 ^a^
Alg/rGO_9_	168.8 ± 12.4 ^b^	53.9 ± 4.2 ^a^	412 ± 45.7 ^b^	42.9 ± 3.2 ^a,b^
Alg/TA_9_	179.4 ± 30.5 ^b^	81.8 ± 8.1 ^b^	298.7 ± 49.3 ^b^	72.4 ± 17.9 ^b^
Alg/rGO_4.5_/TA_9_	170.7 ± 23.16 ^b^	74.8 ± 11.4 ^b^	310.5 ± 57.5 ^b^	64.3 ± 15 ^b^

The letters ^a^ and ^b^ indicate significant statistical differences between samples for *p*-value < 0.05.

## Data Availability

The datasets generated during and/or analyzed during the current study are available from the corresponding author on reasonable request (due to privacy).

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
