# Peer review of "Enhancing Alginate Hydrogels as Possible Wound-Healing Patches: The Synergistic Impact of Reduced Graphene Oxide and Tannins on Mechanical and Adhesive Properties"

_polymers, 2024, doi:10.3390/polym16081081_

Round 1

Reviewer 1 Report

Comments and Suggestions for Authors

This article entitled “Enhancing Alginate Hydrogels as Possible Wound Healing Patches: The Synergistic Impact of Reduced Graphene Oxide and Tannins on Mechanical and Adhesive Properties” describes the synthesis of alginate hydrogels (Alg) using reduced graphene oxide (rGO) and tannins (TA) to evaluate the impact of the concentration of these nanomaterials on mechanical and adhesive, as well as cytotoxicity and wound healing properties. Results reveals improvements in tensile strength, elastic modulus, and toughness upon the incorporation of rGO and TA. This article is really interesting and suitable for publication after the revision. I only have very few comments regarding the manuscript:

1.     Some places grammatical and typographical errors are present in the manuscript. Please rectify the errors. There were several places in this manuscript where the word choices did actively impede understanding. Few places scientific terminology is missing in manuscript. For instance line number 53 and 54 in introduction section. Equation 7 for calculating wound closure rate is missing on page 6, line number 228. Page 6, line number 256 typo is there “tree”. I will suggest again proof reading of manuscript to remove errors.  

2.     Improve the quality of figures. Figure 3 it is suggested to have similar font size on x-axis for all the subpanels.

3.     Shorten the conclusion section.

4.     Although authors have already demonstrated, in vitro wound healing assay for human dermal fibroblast (HDF) cells and obtained satisfactory results. Is it possible to perform a similar scratch assay using pig skin and compare it with the adhesion results obtained earlier?

Author Response

Dear Reviewer,

I send you the improved manuscript and SI, respectively, as well as, the specific answer to the reviewers is attached.

Best regards,

Katherina Fernández

Reviewer 2 Report

Comments and Suggestions for Authors

1) In the Introduction section (lines 87-92) only one method of GO reducing is described - with using dopamine. However, this is not the most common method. Why was this particular method chosen, given that it introduces an additional component (PDA) into the composition of the final hydrogels, the effect of which has not been assessed.

2) The Materials and Methods section is grouped in such a way that it is not entirely clear whether the obtaining of rGO is part of this study or was performed earlier?

3) There is no information about the type and properties of tannins used. What is this product?

4) Lines 170-175. There is duplicate text.

5) Line 229: Equation 7 is missing.

6) Fig. 3a. In fact, the IR spectrum for rGO corresponds to the sum of the spectra of rGO and PDA. The curve signature needs to be adjusted.

7) Fig. 3b is not described in the text. Why does even the introduction of a large amount of TA have no effect on the position of the peaks in the IR spectra of hydrogels?

8) Has there been an assessment of how much PDA remains in the rGO after the reducing stage? Can we consider that the final hydrogels contain not three, but four main components: Alg, rGO, TA + PDA?

9) Fig. 5b. I don’t think it’s correct to connect the experimental points with a line in this case.

10) Fig. 6b. What is the reason for the increase in adhesion force within 10 cycles? Or does this spread fall into a statistical error? If yes, then you need to provide error bars on the graph.

11) Supplementary Materials contain incorrect format of [refs] , highlighted with a yellow marker.

Author Response

(The authors gave the same response as above.)

Reviewer 3 Report

Comments and Suggestions for Authors

1.     Please provide the source and chemical grade of the used materials and chemicals.

2.     Equation 7 is not shown.

3.     Could authors provide magnified SEM images?

Comments on the Quality of English Language

Could authors reorganize paragraphs in the introduction section?

Author Response

(The authors gave the same response as above.)
